# REVEALING SINGLE FRAME BIAS FOR VIDEO-AND-LANGUAGE LEARNING

## ABSTRACT

Training an effective video-and-language model intuitively requires multiple frames as model inputs. However, it is unclear whether using multiple frames is beneficial to downstream tasks, and if yes, whether the performance gain is worth the drastically-increased computation and memory costs resulting from using more frames. In this work, we explore single-frame models for video-and-language learning. On a diverse set of video-and-language tasks (including text-to-video retrieval and video question answering), we show the surprising result that, with large-scale pre-training and a proper frame ensemble strategy at inference time, a single-frame trained model that does not consider temporal information can achieve better performance than existing methods that use multiple frames for training. This result reveals the existence of a strong "static appearance bias" in popular video-and-language datasets. Therefore, to allow for a more comprehensive evaluation of video-and-language models, we propose two new retrieval tasks based on existing fine-grained action recognition datasets that encourage temporal modeling. Code and models will be released upon acceptance.

## 1   INTRODUCTION

Video and language are the two primary signals that constitute much of the world we perceive every day – we observe our surrounding environment with our eyes in the form of continuous visual input (video), and communicate with others via language. Intuitively, this leads one to assume that training an effective video-and-language model should require multiple video frames as input. Standard methods Zhu & Yang (2020); Xu et al. (2021); Li et al. (2020a); Luo et al. (2021) in this area typically use multiple densely sampled frames for training. Recent work Lei et al. (2021) proposes sparse sampling for video-and-language understanding, where it claims that a few sparsely sampled clips are sufficient for learning due to the high redundancy in videos. This technique has shown Lei et al. (2021); Zellers et al. (2021) to be successful in various video-language benchmarks Jang et al. (2017); Xu et al. (2016); Anne Hendricks et al. (2017); Krishna et al. (2017a); Xu et al. (2017); Yu et al. (2018); Lei et al. (2018). However, as demonstrated in Bain et al. (2021); Luo et al. (2021); Lei et al. (2021), training with fewer frames (e.g., a single frame) leads to significantly worse performance compared to their multi-frame counterparts. In contrast, in this work, we show that with proper modeling, single-frame models could achieve competitive performance, hence also revealing "static appearance bias" in popular video-and-language datasets.

We start by building a standard image-language model, with a vision encoder and a language encoder for image and text encoding, followed by a multi-modal encoder with cross-attention for cross-modal fusion. We pre-train the model on large-scale image-text and video-text datasets Chen et al. (2015); Krishna et al. (2017b); Ordonez et al. (2011); Sharma et al. (2018); Changpinyo et al. (2021); Bain et al. (2021). For fine-tuning, we randomly sample a single frame for training, and ensemble multiple uniformly sampled frames per video for making a video-level prediction at inference.

Single-frame predictions are often noisy and inaccurate, as they are made from incomplete information from single-frames without any context (see examples in Figure 5). Due to this issue, single-frame training typically performs significantly worse than multi-frame training Lei et al. (2021); Bain et al. (2021); Luo et al. (2021). Previous work Hendrycks et al. (2019) suggests that pre-training improves model robustness in the face of label corruption for image recognition. Inspired by this, we hypothesize that large-scale pre-training helps mitigate noise from single-frame train-

ing. Our analyses in Section 5 agree with our hypothesis, showing that as we increase pre-training data size, the performance of our single-frame model improves drastically and its gap with a similarly trained multi-frame model is largely eliminated. Besides training, these noisy single-frame predictions also render simple late fusion (e.g., mean-pooling in ClipBERT Lei et al. (2021)) less effective at inference time. To deal with this issue, we propose an early fusion strategy, which takes all frames as model inputs for directly making a more informative video-level prediction. Our analyses show that this early fusion ensemble method outperforms late fusion strategies and also delivers consistently improved performance when more frames are used.

We compare our approach with existing methods on six datasets across two video-language tasks, including text-to-video retrieval (MSRVTT Xu et al. (2016), DiDeMo Anne Hendricks et al. (2017), and ActivityNet Captions Krishna et al. (2017a)) and video question answering (MSRVTT-QA Xu et al. (2017), ActivityNet-QA Yu et al. (2019), and MSRVTT-MC Yu et al. (2018)). Results show that our approach achieves competitive (mostly better) performance than existing methods that use more training frames and more pre-training data, setting new state-of-the-art for multiple tasks. This conclusion holds for short 15-second videos in MSRVTT to 180-second videos in ActivityNet, demonstrating the effectiveness of our single-frame approach in various scenarios.

More importantly, this strong single-frame performance reveals that the current evaluation is biased towards still objects, scenes, etc., while the temporal dynamics seem negligible, which in fact should be important for "true" video-language understanding. To address this issue, we next propose two new tasks that are designed to test models' true temporal modeling ability. Based on the videos and annotations from the find-grained action recognition dataset Something-Something v2 (SSv2) Goyal et al. (2017a), we create two text-to-video retrieval tasks, one that use SSv2's action *template* as text queries, e.g., "Throwing [*something*] in the air and catching it", and another that uses its annotated *label* as text queries, e.g., "Throwing keys in the air and catching it". See examples in Figure 2. This *template* task removes the objects and only keeps the actions, enabling an evaluation that focuses almost solely on temporal modeling. The *label* task, on the other hand, contains both actions and objects, requiring an understanding of both still objects and their motion. Lastly, we present several baselines on these new tasks and show that temporal modeling is essential in achieving high scores.

In summary, our contributions are three-fold: (*i*) We explore single-frame training for video-and-language tasks. While simple, our approach can achieve state-of-the-art performance on a range of datasets, including both text-to-video retrieval and video question answering. Importantly, this result reveals the surprising static appearance bias in these existing datasets. (*ii*) We conduct careful analyses, which show that large-scale pre-training and a proper multi-frame ensemble strategy at inference are the core for single-frame trained models to be successful. (*iii*) We propose two new tasks specifically designed for testing models' ability for find-grained temporal modeling. These two new tasks complement existing benchmarks for a more comprehensive evaluation.

## 2 RELATED WORK

**Vision and Language.** Vision and language learning considers the problem of learning from both visual and textual signals. Depending on their visual input type, methods in this area can be roughly categorized into two types, one with image Anderson et al. (2018); Tan & Bansal (2019); Lu et al. (2019); Chen et al. (2020); Li et al. (2019; 2020b; 2021b; 2022); Radford et al. (2021) and another with video Anne Hendricks et al. (2017); Sun et al. (2019); Zhu & Yang (2020); Xu et al. (2021); Li et al. (2020a); Lei et al. (2021); Zellers et al. (2021); Bain et al. (2021); Lin et al. (2021). Standard video-and-language methods Zhu & Yang (2020); Xu et al. (2021); Li et al. (2020a); Lei et al. (2021); Zellers et al. (2021); Luo et al. (2021) are typically trained with multiple video frames. This multi-frame training strategy has been the norm and is shown to work well across various datasets Xu et al. (2016); Anne Hendricks et al. (2017); Krishna et al. (2017a); Jang et al. (2017); Xu et al. (2017); Lei et al. (2018; 2020). Unlike previous work that uses multiple frames for training, we explore single-frame training (i.e., similar to training an image-text model) and show it achieves strong performance on existing video-text benchmarks. Concurrent work Buch et al. (2022) proposes a new module, atemporal probe, for selecting the best single-frame as inputs to a trained image-text model during inference; whereas we utilize multiple uniformly sampled frames and study more effective ways of ensembling information from multiple frames.

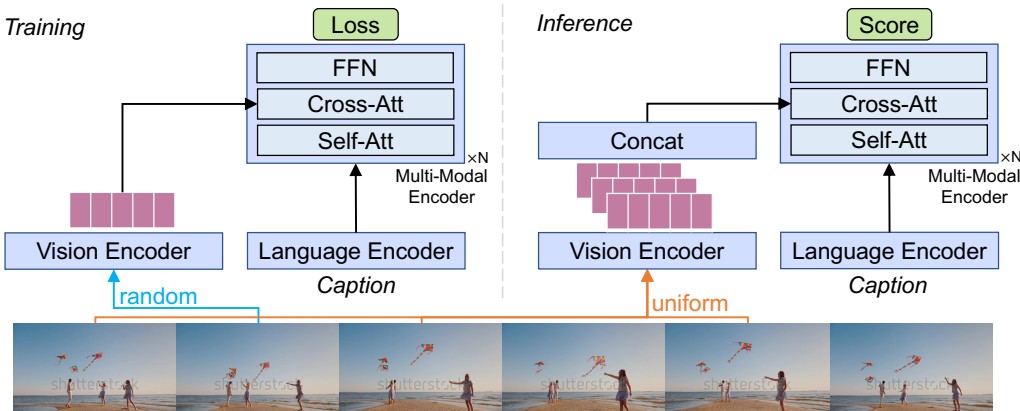

*Caption:* A group of people play kites together on the beach.

Figure 1: SINGULARITY model overview. During training, we randomly sample a single frame as input, and make a video level prediction based on the information from this single frame along with its paired text input. During inference, we uniformly sample multiple frames, and early fuse their encoded image-level representations as input to the multi-modal encoder. See details in Section 3.

**Dataset Bias.** Biases are prevalent in datasets Goyal et al. (2017b); Gururangan et al. (2018); Li et al. (2018); Escorcia et al. (2019); Zellers et al. (2019); Lei et al. (2020). For example, Zhang et al. Zhang et al. (2016) pointed out that blindly answering "yes" to yes/no questions in VQA Antol et al. (2015) without looking at their corresponding images results in an accuracy of 87%; Li et al. Li et al. (2018) discovered that many video action recognition datasets, such as Kinetics Kay et al. (2017) and UCF-101 Soomro et al. (2012), have a strong static representation, where a linear classifier trained on static appearance (e.g., object, scene, and people) representations achieves much higher performance than chance. In this work, we find similar static appearance bias exists in popular video-language datasets Xu et al. (2016); Anne Hendricks et al. (2017); Krishna et al. (2017a); Xu et al. (2017); Yu et al. (2018; 2019), in which our models trained with single frames could achieve surprisingly good performance, even compared to models that perform explicit temporal modeling. When datasets are biased, they provide incorrect indications of the models' ability. To allow for a more comprehensive evaluation, we propose two new tasks based on an existing action recognition dataset SSv2 Goyal et al. (2017a) to test the true temporal modeling ability of models.

## 3 METHODS

**Model Architecture.** Figure 1 shows an overview of our model (dubbed SINGULARITY). It consists of 3 main components, a vision encoder $\mathcal{F}_v$, a language encoder $\mathcal{F}_l$, and a multi-modal encoder $\mathcal{H}$. The vision encoder is an image-level visual backbone model, such as ViT Dosovitskiy et al. (2020). The language encoder is an arbitrary language model such as BERT Devlin et al. (2019). For the multi-modal encoder, we use a transformer encoder Vaswani et al. (2017), in which each layer contains a self-attention, a cross-attention, and a feed-forward network (FFN). The cross-attention layer is used to gather information from encoded visual representations using the text as key, similar to recent work Jaegle et al. (2021; 2022); Li et al. (2021b; 2022).

We denote a video $V$ contains $T$ frames as $V=[f_1, f_2, ..., f_T]$, its paired text as $S$. During training, we randomly sample a single frame $f_t$ from $V$ as model input, where $t \in \{1, ..., T\}$. Its encoded representation can be written as $\mathcal{F}_v(f_t) \in \mathbb{R}^{L_v \times D}$. For text, the encoded representation is $\mathcal{F}_l(S) \in \mathbb{R}^{L_l \times D}$. $L_v$ and $L_l$ are encoded sequence lengths, $D$ is hidden size. We next make a prediction $p$ as:

$$p = \mathcal{H}(\ \mathcal{F}_l(S)\ ,\ \mathcal{F}_v(f_t)\ ), \tag{1}$$

Q, K, V for self-att; Q for cross-att ↑     ↑ K, V for cross-att

where Q, K, V denote the query, key, and value matrices of self- and cross-attention Vaswani et al. (2017). We calculate loss based on this prediction. During inference, we uniformly sample $T_{test}$ frames $\{f_{\tau_i}\}_{i=1}^{T_{test}}$. Each frame is encoded separately, and their encoded representations are concate-

nated as inputs to the multi-modal encoder to get a video-level prediction score:

$$p = \mathcal{H}(\ \mathcal{F}_l(S)\ ,\ [\mathcal{F}_v(f_{\tau_1}); ...; \mathcal{F}_v(f_{\tau_{T_{test}}})]\ ), \tag{2}$$

where $[;]$ denotes concatenation, and $[\mathcal{F}_v(f_{\tau_1}); ...; \mathcal{F}_v(f_{\tau_{T_{test}}})] \in \mathbb{R}^{(T_{test} \times L_v) \times D}$. This early fusion design allows our model to make an informed prediction given full context. In ClipBERT Lei et al. (2021), an alternative late fusion design is used: scores are computed for each frame separately, and video-level score is obtained via a manually designed aggregation function $\mathcal{G}$ (e.g., mean-pooling):

$$p = \mathcal{G}(p_{\tau_1}, p_{\tau_2}, p_{\tau_{T_{test}}}); \quad p_{\tau_i} = \mathcal{H}(\ \mathcal{F}_l(S)\ ,\ \mathcal{F}_v(f_{\tau_i})\ ). \tag{3}$$

Since the predictions in late fusion are made with incomplete information from individual frames, they can be quite noisy. In Section 5, we provide a detailed comparison w.r.t. these different frame ensemble methods and show that early fusion consistently outperforms late fusion.

**Pre-Training Objectives.** The model is trained with 3 losses: (*i*) Vision-Text Contrastive: a contrastive loss that aligns the pooled vision and text representations from the vision and language encoders. (*ii*) Masked Language Modeling (MLM) Devlin et al. (2019): predicting masked tokens from their text and visual context, with multi-modal encoder. (*iii*) Vision-Text Matching: predicting the matching score of a vision-text pair with multi-modal encoder. These losses have shown to be effective in learning multi-modal representations Tan & Bansal (2019); Chen et al. (2020); Li et al. (2021a;b); Lei et al. (2021); Radford et al. (2021). More details are in Appendix.

**Implementation Details.** As our model trains with single frames, in addition to video-text data, it can also utilize image-text data for pre-training. For image-text data, we use a combination of COCO Chen et al. (2015), Visual Genome (VG) Krishna et al. (2017b), SBU Captions Ordonez et al. (2011), CC3M Sharma et al. (2018), and CC12M Changpinyo et al. (2021). For video-text data, we use WebVid Bain et al. (2021). Note that, even for video-text data, we only sample a single frame from the whole video for training. We pre-train the model on two different subsets of the datasets: (*i*) 5M corpus that contains 5.44M images and videos from CC3M+WebVid, and (*ii*) 17M corpus that contains 17.28M images and videos from all the datasets above.

Our model is implemented in PyTorch Paszke et al. (2019). The vision encoder is initialized using the BEiT$_{\text{BASE}}$ Bao et al. (2021) model pre-trained on ImageNet-21K Deng et al. (2009). The text encoder is initialized from the first 9 layers of BERT$_{\text{BASE}}$ Devlin et al. (2019). The multi-modal encoder is initialized from the last 3 layers of the same BERT$_{\text{BASE}}$ model, though its cross-attention layers are randomly initialized. We optimize the model for 10 epochs using AdamW Loshchilov & Hutter (2019) optimizer with an initial learning rate of 1e-4. We warm up the learning rate in the first epoch followed by cosine decay Loshchilov & Hutter (2017) to 1e-6 during the rest of the training. Mixed precision is used for faster training. The batch size is set to 128 per GPU, and we train the model on 3 NVIDIA A100 GPUs with input image size 224×224. We perform basic augmentations: random resize, crop, and flip to the frames/images during training. This pre-training takes around 1 day on the 5M corpus, and 4 days on the 17M corpus. Our pre-training is quite efficient compared to other similar work, e.g., 10 epochs' pre-training in AlignPrompt Li et al. (2021a) takes 3 days on the same 5M corpus using 16 A100 GPUs, this amounts to 16× computation cost of our pre-training.

## 4 EXPERIMENTS AND RESULTS ON EXISTING DATASETS

### 4.1 DOWNSTREAM TASK SETUP

**Text-to-Video Retrieval.** Given a text query, the goal of this task is to retrieve relevant videos from a large collection of videos. We evaluate our model on the following datasets: (*i*) **MSRVTT** Xu et al. (2016) contains 10K YouTube videos, each paired with 20 captions. We follow Yu et al. (2018); Lei et al. (2021) to use the 7K train+val videos for training, and report results on the 1K test set. (*ii*) **DiDeMo** Anne Hendricks et al. (2017) contains 10K Flickr videos with 41K captions. We use standard train/val/test splits. (*iii*) **ActivityNet Captions** Krishna et al. (2017a) contains 20K YouTube videos with 100K captions. We use the train split with 10K videos for training, and we report results on the widely used val1 split, with 4.9K videos. For MSRVTT, we evaluate standard text-to-video retrieval. For DiDeMo and ActivityNet Captions, we evaluate paragraph-to-video retrieval Liu et al. (2020); Lei et al. (2021); Luo et al. (2021), where the text captions in the

Table 1: Comparison to existing methods on text-to-video retrieval. *#PT* denotes the number of images and or videos used in cross-modal pre-training. *#Train Frame* denotes the number of frames used at each training step during fine-tuning. For models that use different number of frames for different datasets, we list them together with a separator "/". We gray out methods that use significantly more pre-training data for a fair comparison. The 136M corpus is from HowTo100M Miech et al. (2019), 0.2M refers to COCO+VG data, 138M is the combination of HowTo100M and WebVid, 400M is the private image-text data used in CLIP Radford et al. (2021).

| Method | #PT | #Train Frame | MSRVTT | | | DiDeMo | | | ActivityNet Cap | | |
|---|---|---|---|---|---|---|---|---|---|---|---|
| | | | R1 | R5 | R10 | R1 | R5 | R10 | R1 | R5 | R10 |
| HERO (Li et al., 2020a) | 136M | 310 | 20.5 | 47.6 | 60.9 | - | - | - | - | - | - |
| ClipBERT (Lei et al., 2021) | 0.2M | 16/16/8 | 22.0 | 46.8 | 59.9 | 20.4 | 48.0 | 60.8 | 21.3 | 49.0 | 63.5 |
| VideoCLIP (Xu et al., 2021) | 136M | 960 | 30.9 | 55.4 | 66.8 | - | - | - | - | - | - |
| Frozen (Bain et al., 2021) | 5M | 4 | 31.0 | 59.5 | 70.5 | 31.0 | 59.8 | 72.4 | - | - | - |
| AlignPrompt (Li et al., 2021a) | 5M | 8 | 33.9 | 60.7 | 73.2 | 35.9 | 67.5 | 78.8 | - | - | - |
| All-in-one (Wang et al., 2022) | 138M | 9 | 34.4 | 65.4 | 75.8 | 32.7 | 61.4 | 73.5 | 22.4 | 53.7 | 67.7 |
| CLIP4Clip (Luo et al., 2021) | 400M | 12/64/64 | 42.0 | 68.6 | 78.7 | 42.8 | 68.5 | 79.2 | 40.5 | 72.4 | 98.2 |
| SINGULARITY | 5M | 1 | 36.8 | 65.9 | 75.5 | 47.4 | 75.2 | 84.0 | 43.0 | 70.6 | 81.3 |
| SINGULARITY | 17M | 1 | **41.5** | **68.7** | **77.0** | **53.9** | **79.4** | **86.9** | **47.1** | **75.5** | **85.5** |

same video are concatenated as a single paragraph-level text for retrieval. We report performance using recall at K (R@K).

For fine-tuning, we use the same architecture as pre-training, except that MLM loss is removed. We use an initial learning rate of 1e-5 with cosine decay to 1e-6. We use a batch size of 32, and train the model for 5 epochs for MSRVTT, 10 epochs for DiDeMo and ActivityNet Captions. During training, we use a single frame per video. During testing, we use 12 frames per video for MSRVTT and DiDeMo, and 32 frames for ActivityNet Captions since it has longer videos. On a single A100, this fine-tuning takes around 1.5 hours for MSRVTT, 0.5 hours for ActivityNet Captions or DiDeMo.

**Video Question Answering.** Given a video (often with a text question), this task requires generating an answer to the question or selecting the most suitable answer from a set of candidates. (*i*) **MSRVTT-QA** Xu et al. (2017) contains 244K open-ended questions on 10K MSRVTT videos. (*ii*) **ActivityNet-QA** Yu et al. (2019) contains 58K open-ended questions on 5.8K sampled videos from ActivityNet Caba Heilbron et al. (2015). (*iii*) **MSRVTT-MC** Yu et al. (2018) is a multiple-choice task that requires selecting the matched caption from a set of 5 candidate captions for each video (3K videos from MSRVTT). We use standard train/val/test splits for the three tasks, and report accuracy.

For open-ended QA, we add an extra multi-modal decoder (initialized from pre-trained multi-modal encoder) that takes in multi-modal encoder outputs as cross-attention inputs, and decodes answer text with "`[CLS]`" as the start token (see details in Appendix). We use an initial learning rate of 1e-5, and warm up the learning rate in the first half epoch, followed by cosine decay to 1e-6. We use a batch size of 32, and train the model for 10 epochs. On a single A100 GPU, this fine-tuning takes around 4 hours for MSRVTT-QA, and 1 hour for ActivityNet-QA. We use a single frame per video for training, 12 frames for testing. For MSRVTT-MC, we follow Lei et al. (2021) to use the model trained on MSRVTT retrieval, and select the option with the highest retrieval score as the prediction.

For all downstream tasks, we use the same input image size 224×224 and image augmentations as in pre-training. During inference, we resize the input video frames to 224×224.

## 4.2 COMPARISON TO STATE-OF-THE-ART ON EXISTING DATASETS

**Text-to-Video Retrieval Results.** In Table 1, we compare SINGULARITY with existing methods on text-to-video retrieval. Across all the datasets, SINGULARITY (5M) achieves better performance compared to methods trained on similar amounts of data, while using only single frames for training. On DiDeMo and ActivityNet Captions, SINGULARITY (5M) outperforms all previous work, including many that pre-train on significantly larger amounts of data, e.g., 400M image-text pairs

Table 2: Comparison to existing methods on video question answering. The 69M corpus is the 69M video questions in Yang et al. (2021), 180M refers to the 180M YouTube clip-text pairs in YT-Temporal-180M Zellers et al. (2021).

| Method | #PT | #Train Frame | MSRVTT-QA | ActivityNet-QA | MSRVTT-MC |
|---|---|---|---|---|---|
| ClipBERT (Lei et al., 2021) | 0.2M | 16 | 37.4 | - | 88.2 |
| AlignPrompt (Li et al., 2021a) | 5M | 16 | 42.1 | - | - |
| JustAsk (Yang et al., 2021) | 69M | 640 | 41.5 | 38.9 | - |
| MERLOT (Zellers et al., 2021) | 180M | 5 | 43.1 | 41.4 | 90.9 |
| VideoCLIP (Xu et al., 2021) | 136M | 960 | - | - | 92.1 |
| All-in-one (Wang et al., 2022) | 138M | 9 | 44.3 | - | 92.0 |
| SINGULARITY | 5M | 1 | 42.7 | 41.8 | 92.0 |
| SINGULARITY | 17M | 1 | **43.5** | **43.1** | **92.1** |

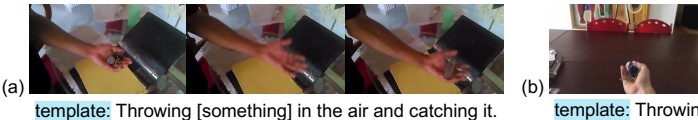

Figure 2: SSv2 examples. For each video, we show 3 temporally-ordered frames with their *template* and *label* annotations. Based on these annotations, we propose two new retrieval tasks, using "template" and "label" as text queries, respectively.

in CLIP4Clip Luo et al. (2021), or 136M video-text pairs in VideoCLIP Xu et al. (2021) compared to 5M image-text and video-text pairs in SINGULARITY. We also note that our model is trained with single frames, while previous work uses many more frames, e.g., 64 frames in CLIP4Clip or 8 frames in AlignPrompt Li et al. (2021a). When trained with a larger amount of data (17M), we notice a further performance boost for our model, demonstrating that SINGULARITY benefits from large-scale pre-training.

**Video QA Results.** Table 2 compares SINGULARITY with existing methods on video question answering. We notice SINGULARITY (5M) achieves competitive performance with previous work even when using two orders of magnitude smaller pre-training data, e.g., 180M video-text pairs in MERLOT Zellers et al. (2021) *vs.* 5M image-text and video-text pairs. Our method also surpasses the strong video QA model JustAsk Yang et al. (2021), which is specifically designed for video QA and is pre-trained on 69M video QA pairs. When pre-trained with more data, our model performance further improves. These comparisons show the effectiveness of our single-frame approach.

Beyond what are present in the main text, we also provide additional results in Appendix: (*i*) SINGULARITY-temporal (introduced in Section 4.3) results on retrieval and QA; (*ii*) zero-shot retrieval; (*iii*) image-text retrieval; (*iv*) VQA Antol et al. (2015), etc.

## 4.3 NEW TASKS THAT REQUIRE TEMPORAL MODELING

In the previous section, we revealed the interesting observation that popular video-language datasets have strong static appearance biases – enabling our model that uses only a single frame per video at each training step to achieve competitive performance compared to state-of-the-art models that digest multiple temporally-ordered frames. The biased evaluation on these datasets favors models that are strong in recognizing static concepts, and does not provide a good indicator of whether these models are capable of recognizing fine-grained temporal relationships between neighboring frames.

Hence, to address this issue, we propose two new datasets that complement existing datasets for a more comprehensive evaluation of video-and-language methods. We draw inspiration from the video action recognition community, and transform the temporally-heavy action recognition dataset Something-Something v2 (SSv2) Goyal et al. (2017a) into video-and-language datasets. In Figure 2,

Table 3: Comparison to existing methods on SSv2 tasks. * The training of Frozen on the SSv2-label retrieval task fails to converge despite our best efforts in tuning the model.

| Method | #PT | #Train Frame | SSv2-label | | | SSv2-template | | |
|---|---|---|---|---|---|---|---|---|
| | | | R1 | R5 | R10 | R1 | R5 | R10 |
| Frozen (Bain et al., 2021)* | 5M | 4 | - | - | - | 52.9 | 94.8 | **99.4** |
| CLIP4Clip (Luo et al., 2021) | 400M | 12 | 43.1 | 71.4 | 80.7 | 77.0 | 96.6 | 98.3 |
| SINGULARITY | 5M | 1 | 36.4 | 64.9 | 75.4 | 42.0 | 86.2 | 94.3 |
| SINGULARITY-temporal | 5M | 4 | 44.1 | 73.5 | 82.2 | 77.0 | 98.9 | **99.4** |
| SINGULARITY-temporal | 17M | 4 | **47.4** | **75.9** | **84.0** | **77.6** | 96.0 | 98.9 |

we show SSv2 examples. A unique property of the SSv2 dataset is that the videos often require fine-grained temporal modeling to correctly predict their action classes. For example, to match the videos and their action classes (*template*) in Figure 2(a-b), one has to look at multiple temporally ordered frames. Based on SSv2 videos and annotations, we define two text-to-video retrieval tasks:

- **SSv2-Template Retrieval**: We use the 174 templates (e.g., "Throwing [*something*] in the air and catching it") in SSv2 as the text queries to retrieve videos. We use 168,913 SSv2 training videos for training. As ground-truth annotations for test videos are not available, we use validation videos: we sample 12 videos for each template, with a total of 2,088 videos for testing.

- **SSv2-Label Retrieval**: We use the annotated labels (e.g., "Throwing keys in the air and catching it") in SSv2 as text queries to retrieve videos. We follow the same split in the template retrieval task, with 168,913 videos for training, and 2,088 videos for testing.

Since no objects are present in the text queries of the template retrieval task, it requires a deeper understanding of the actions than in the label retrieval task, while the label retrieval task provides a more comprehensive evaluation of both static and temporal understanding.

**Experiments.** We use Frozen Bain et al. (2021) and CLIP4Clip (seqTransf version) Luo et al. (2021) as baselines. Frozen uses a space-time transformer, CLIP4Clip is an extension based on the CLIP Radford et al. (2021) with an extra 4-layer temporal transformer encoder. We report performance using standard text-to-video retrieval metrics R@K. For our model, in addition to the single-frame version, we build a multi-frame variant, SINGULARITY-temporal. Specifically, we add a two-layer temporal transformer encoder following the vision encoder, and use its outputs as inputs to the multi-modal encoder (see details in Appendix). From a single-frame pre-trained checkpoint (5M or 17M), we perform a 2nd stage video pre-training with 4 frames using WebVid videos for SINGULARITY-temporal. We use an initial learning rate of 5e-5, and train the model for 5 epochs.

The results are shown in Table 3. Compared to Frozen and CLIP4Clip, while SINGULARITY shows competitive performance on existing benchmarks (see Table 1), it underperforms these methods on the two temporally-heavy tasks by a large margin. For example, SINGULARITY (5M) underperforms the 4-frame Frozen model by 10.9 for SSv2-template retrieval R1, though it shows a 16.4 improvement for DiDeMo R1, and 5.8 for MSRVTT R1. This is a good sign as it shows that the new tasks cannot be solved by models exploiting static appearance biases. On the other hand, after adding the 2-layer temporal encoder, the 4-frame SINGULARITY-temporal model gets a significant performance boost from the single-frame model, surpassing the baseline methods. When using more pre-training data (5M→17M), we notice a good performance gain for SSv2-label, while the performance on SSv2-template stays similar. These observations indicate that the SSv2-label task requires both static and temporal modeling, and enhancing either will improve the task performance. For SSv2-template, as no objects exist in its text queries, it requires mostly temporal modeling.

## 5 ANALYSIS

**Frames Ensemble Strategy.** Our model is trained with a single-frame regime, and it uses multiple frames covering the full video at inference. As shown in Figure 3a (*concat*), encoded video frames are concatenated as input to the multi-modal encoder's cross-attention layer for making a video-level prediction. A naive alternative is to compute the prediction score for each frame separately

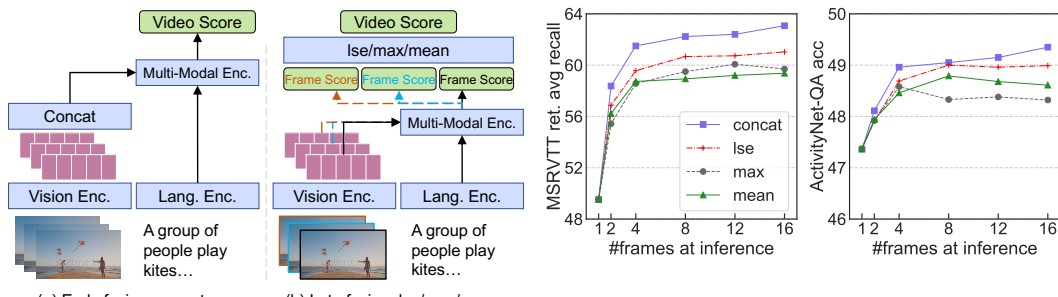

Figure 3: Comparison of frame ensemble strategies at inference. *concat* is our early fusion strategy, *lse*, *max*, *mean* are the late fusion strategies studied in ClipBERT Lei et al. (2021).

Figure 4: Impact of frame ensemble strategy. Retrieval performance is shown as *avg recall*, i.e., average of R@{1,5,10}. We use the same fine-tuned checkpoint for each task, thus the results difference only comes from inference strategies.

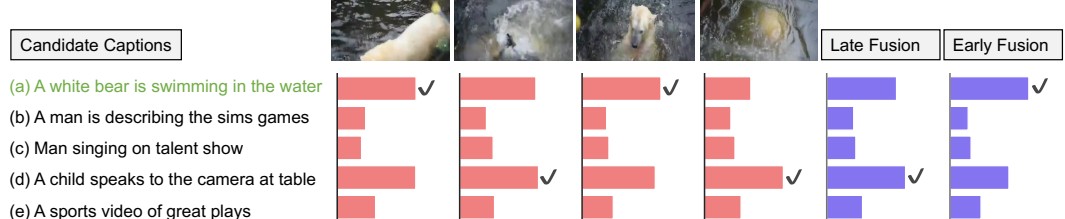

Figure 5: Prediction score distribution for a MSRVTT-MC example. We show frame-level score distribution for each frame, and video-level score distribution for late fusion (we use *mean* as an example) and our early fusion (*concat*). The highest score for each prediction is indicated by ✓, the correct answer is highlighted in green. Single-frame predictions are often inaccurate, unstable and they fluctuate across the frames. Late fusion can be biased by inaccurate but high confidence frame predictions, e.g., the late fusion prediction is biased towards the 4th frame prediction.

(Figure 3b), and then aggregate these frame-level scores together to get a video-level score using an aggregation function, such as LogSumExp (*lse*), *max*-pooling and *mean*-pooling. This simple late fusion strategy has shown to be successful for video-and-language Lei et al. (2021) and video action recognition methods Bertasius et al. (2021); Carreira & Zisserman (2017); Wang et al. (2016).

In Figure 4, we compare these different frame ensemble strategies, with varying number of frames at inference. From the comparison, we can draw the following conclusions: ($i$) Our early fusion strategy (*concat*) shows a significant gain over the three late fusion strategies (*lse*, *max*, *mean*) for both MSRVTT retrieval and ActivityNet-QA, demonstrating the importance of considering the whole video when making the predictions. ($ii$) In general, for all ensemble strategies, using more frames at inference improves model performance. However, for the late fusion strategies, sometimes using more frames hurts performance, e.g., for ActivityNet-QA, inference with over 4 frames underperforms that with 4 frames for max-pooling. This observation agrees with the MSRVTT-QA results in ClipBERT Lei et al. (2021). In contrast, early fusion delivers consistently improved performance when more frames are used. Overall, we hypothesize that the low and unstable performance of late fusion is because its video-level prediction is obtained via aggregating frame-level predictions, while these frame-level predictions can be inaccurate and unstable (see example in Figure 5) – as they are separately predicted using incomplete information within each frame, ignoring their context.

**Pre-Training Data Size.** In Figure 6, we study the effect of cross-modal pre-training data size for both the single-frame and the multi-frame model. We show downstream fine-tuning performance under 4 different pre-training data setups: no cross-modal pre-training (0M), pre-train on WebVid (2.49M videos), on 5M corpus (5.44M images+videos), or on 17M corpus (17.28M images+videos).

We obsereve that both 1-frame and 4-frame model greatly benefit from large-scale pre-training. When comparing the two models, an interesting observation is that, as the pre-training data size increases, the performance gap between the 1-frame and the 4-frame model decreases almost mono-

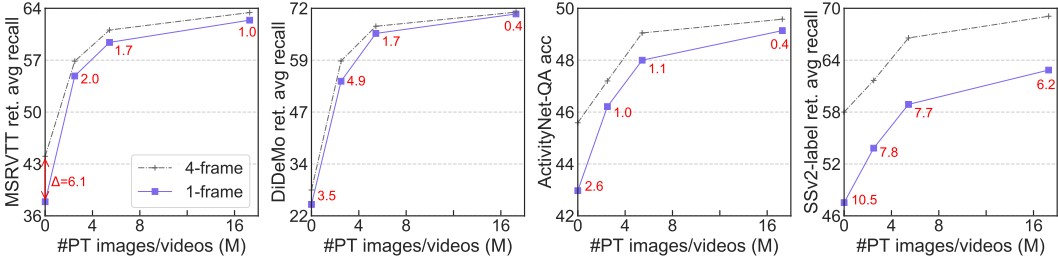

Figure 6: Model performance as a function of pre-training data size, for SINGULARITY (1-frame) and SINGULARITY-temporal (4-frame). The performance differences between the two models in each pre-training setup is also annotated, e.g., the average recall on MSRVTT retrieval for the two models without pre-training are 37.9 and 44.0, respectively, with Δ=6.1. In general, as pre-training data size increases, the performance gap between the two models decreases.

tonically. This phenomenon suggests that, when pre-trained on a sufficient amount of data, the performance of models trained with single frames might be very close to models trained with multiple frames. Though there can be exceptions for tasks that require fine-grained temporal modeling, such as SSv2-label retrieval, where multi-frame modeling is necessary.

One possible explanation is that single-frame training is noisier than multi-frame training – due to incomplete context and random sampling, single-frame predictions are often inaccurate and less stable than multi-frame predictions, and pre-training is helpful Hendrycks et al. (2019) in this case. Meanwhile, single-frame training requires the model to extract all information from a single frame while a multi-frame model could rely on rich sources from multiple frames. Therefore, for downstream tasks, it is essential for the single-frame model to initialize from a strong pre-trained model.

**Training Efficiency.** A core advantage of single-frame training is its training efficiency. In Section 3, we discussed our pre-training cost is only 1/16 of a recent video-language model Li et al. (2021a). In Figure 7 we compare the training time and task performance of various models. We note our model (1-frame, SINGULARITY, 17M) trains much faster than the baselines (2.8× for 4-frame Frozen, 8.5× for 64-frame CLIP4Clip) while showing significantly better performance. Besides, it is also more memory efficient, i.e., its maximum allowed batch size on a single GPU is 190 while only 50 for Frozen. Experiments conducted on a single RTX A6000 GPU with 48GB memory, training time is averaged over 8,394 DiDeMo training examples. In Appendix, we show additional comparisons of various retrieval methods in terms of inference GFLOPs and the number of model parameters.

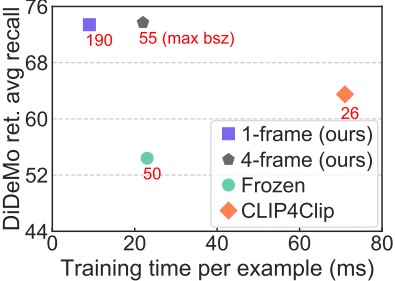

Figure 7: Comparison of training time and downstream task performance. The maximum allowed batch size is labeled besides each model as a reference.

## 6 CONCLUSION

In this work, we explore single-frame training for video-and-language learning. We find that, with sufficient pre-training data and a proper frame ensemble strategy at inference, our model trained with a single frame achieves surprisingly good performance on various video-text tasks, including text-to-video retrieval and video question answering. While these results show the potential of using single-frame training for various video-text tasks, it also reveals that current benchmarks are biased towards static objects and scenes, etc. To address this issue, we propose two new tasks designed to test models' true temporal modeling ability and build several baseline methods for these new tasks. We hope these new tasks can complement existing benchmarks for a more comprehensive video-and-language understanding.

**Societal Impact.** Similar to many data-driven methods, the predictions from our system reflect the distribution of data on which it is trained on, and these predictions can be inaccurate and biased by the data. Therefore, users should not completely rely on the system for making real-world decisions.

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

## A  APPENDIX

In Section A.2, we show details of our open-ended QA model and SINGULARITY-temporal model, as well as pre-training objectives. In Section A.3, we show more experimental details, such as SINGULARITY-temporal results on existing datasets, SINGULARITY zero-shot results, impact of image size, and results on image-text tasks such as text-to-image retrieval tasks Flickr30K Young et al. (2014), COCO Chen et al. (2015) and image question answering task VQA Antol et al. (2015). In addition, we also show hyper-parameters and more experimental setups in this section. In Section A.4, we show more dataset details.

### A.1  AUTHOR RESPONSE

**Memory and Time Cost of Frame Ensemble Strategies.** In Sec.5, we discussed that our simple early fusion based frame ensemble strategy (*concat*) achieves the best performance for both MSRVTT retrieval and ActivityNet-QA tasks across different number of inference frames. In this section, we continue to compare its memory and computation time cost w.r.t. other frame ensemble strategies. For both tasks, our early fusion strategy (*concat*) achieves the better performance than late fusion strategies (*lse*, *max*, *mean*) while also runs faster. For memory cost, *concat* uses more memory for MSRVTT retrieval, but fewer memory for the ANet-QA. Overall, the early fusion approach is preferred in most cases due to its better accuracy and faster run time.

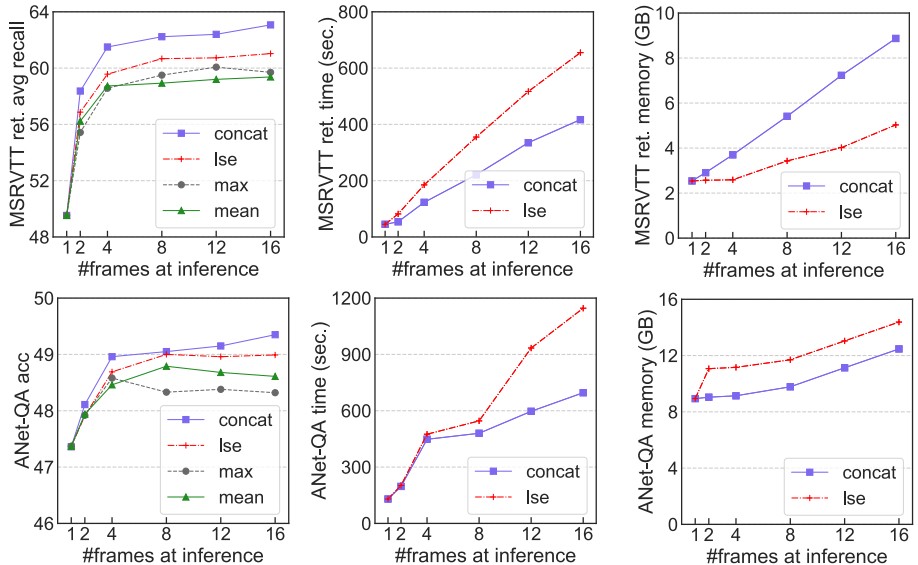

Figure 8: Impact of frame ensemble strategy. Retrieval performance is shown as *avg recall*, i.e., average of R@{1,5,10}. *Top* row shows the performance, time and memory comparisons for MSRVTT retrieval task, while *bottom* shows the same comparisons for ActivityNet-QA (ANet-QA). We use the same fine-tuned checkpoint for each task, thus the results difference only comes from inference strategies. We measure time and memory cost by running the models on the task-specific test splits. Since the three late fusion strategies (*lse*, *max*, *mean*) have similar memory and time costs, we only keep *lse* in the figures.

## A.2 Additional Modeling Details

**Open-ended QA model.** Figure 9a shows a graphic overview of the model architecture for open-ended video question answering. Following previous work Cho et al. (2021); Li et al. (2021b), we formulate this task as text generation instead of classification. Based on the base model described in main text, we add an extra multi-modal decoder that takes in multi-modal encoder outputs as cross-attention inputs, and decodes answer text with "[CLS]" as the start token. This decoder has the exact same architecture as the multi-modal encoder. We initialize its weight using the pre-trained multi-modal encoder.

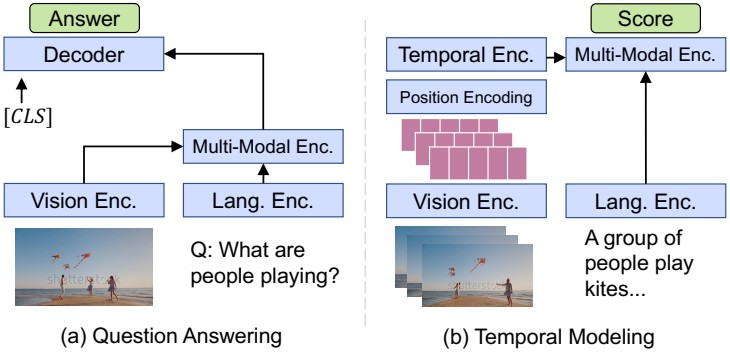

Figure 9: SINGULARITY model variants for video question answering and temporal modeling (i.e., SINGULARITY-temporal). The horizontal arrows indicate cross-attention inputs, while the vertical arrows indicate self-attention inputs.

**SINGULARITY-temporal.** Figure 9b shows a graphic overview of the model architecture for temporal modeling, this model is also referred to as SINGULARITY-temporal. Given multiple video frames $\{f_{\tau_i}\}_{i=1}^{T_{train}}$ as input, the model firstly encode each frame into their visual representations $\{\mathcal{F}_v(f_{\tau_i})\}$

with the vision encoder $\mathcal{F}_v$, where $\mathcal{F}_v(f_{\tau_i}) \in \mathbb{R}^{L_v \times D}$. Next, we add temporal position encoding to each frame to indicate their temporal order. This temporal position encoding is learned from scratch and is initialized as zeros. For brevity, we omit this encoding in the formulation. These frame-level representations are concatenated together as input to the temporal encoder $\mathcal{T}$, and we feed temporal encoder outputs to the multi-modal encoder's cross-attention layer for making a prediction $p$:

$$p = \mathcal{H}(\ \mathcal{F}_l(S)\ ,\ \mathcal{T}([\mathcal{F}_v(f_{\tau_1}); ...; \mathcal{F}_v(f_{\tau_{T_{train}}})]) \ ),\tag{4}$$

Q, K, V for self-att; Q for cross-att ↑           ↑ K, V for cross-att

where $[;]$ denotes concatenation, and $[\mathcal{F}_v(f_{\tau_1}); ...; \mathcal{F}_v(f_{\tau_{T_{train}}})] \in \mathbb{R}^{(T_{train} \times L_v) \times D}$. During inference, when $T_{test}$ frames are used as inputs to the model and $T_{test} > T_{train}$, we interpolate the temporal position encoding to allow for extended temporal length. This is similar to spatial position encoding interpolation in Touvron et al. (2021).

**Pre-Training Objectives.** During pre-training, we optimize the model with three standard vision-and-language objectives, Vision-Text Contrastive (VTC), Masked Language Modeling (MLM) Devlin et al. (2019), and Vision-Text Matching. We explain them in detail below.

(*i*) **Vision-Text Contrastive** (VTC) loss aims to aligns paired vision and language embeddings. Given the encoded vision embedding $\mathcal{F}_v(f_{i,t})$, we use a projection head (with pooling) $\phi_v$ to project the embedding sequence into a vector representation $\phi_v(\mathcal{F}_v(f_{i,t})) \in \mathbb{R}^D$. Here $f_{i,t}$ is the $t$-th frame in the $i$-th video in the training set, and $t$ is randomly sampled from all available frames in this video. For brevity, we omit the subscript $t$ and use $f_i$ to denote a randomly sampled frame from the $i$-th video during the rest of the discussion. Similarly, we have $\phi_l(\mathcal{F}_l(S_j)) \in \mathbb{R}^D$ for the $j$-th sentence. The similarity score $s_{i,j}$ of the video and text pair is defined as their dot product:

$$s_{i,j} = \phi_v(\mathcal{F}_v(f_i))^T \phi_l(\mathcal{F}_l(S_j))\tag{5}$$

We apply a contrastive loss to encourage the alignment between paired vision-language embeddings:

$$p_i^v = \frac{\exp(s_{i,i}/\tau)}{\sum_j \exp(s_{i,j}/\tau)}, \ \ p_i^l = \frac{\exp(s_{i,i}/\tau)}{\sum_j \exp(s_{j,i}/\tau)}, \mathcal{L}_{vtc} = -\sum_{i=1}^n (\log p_i^v + \log p_i^l),\tag{6}$$

where $\tau$ is a learned temperature parameter, and it is initialized as 0.07 following CLIP Radford et al. (2021). $n$ is the total number of examples in the training set.

(*ii*) **Masked Language Modeling** (MLM) loss, or more precisely, Vision Conditioned Masked Language Modeling loss, aims to predict masked text tokens from their (masked) textual context as well as the visual context. This loss is applied at the last layer of the multi-modal encoder, and we follow the exact formulation in BERT Devlin et al. (2019), except that we add additional vision inputs and use a higher mask ratio of 50%.

(*iii*) **Vision-Text Matching** (VTM) loss works towards the same goal as the VTC loss – encouraging the alignment between paired vision and language inputs. It uses the [CLS] output from the multi-modal encoder for binary classification – whether the input vision and language pair match or not. To make the training more effective, we also leverage hard negative sampling Li et al. (2021b); Chen et al. (2020) to sample more informative negatives within the batch for VTM.

## A.3 ADDITIONAL EXPERIMENTS

**Analysis Setup.** For all ablation studies, we report results on validation splits for the datasets if available. For example, we use validation splits for DiDeMo retrieval and ActivityNet-QA, and we use the test split for MSRVTT retrieval, val1 split for ActivityNet Captions retrieval, and test split for SSv2-label. For retrieval tasks, we use the average recall, which is the average score of R@{1,5,10}) to more holistically compare the model performance. For QA tasks, we use accuracy.

**SINGULARITY-temporal Results on Existing Datasets.** In Table 4 and Table 5 we show results of SINGULARITY-temporal on existing text-to-video retrieval and video question answering datasets. In general, the 4-frame model SINGULARITY-temporal improves upon the 1-frame model SINGULARITY, but the performance gap is relatively small, especially considering the greatly increased memory and computation cost (discussed in main text) of using 4 frames.

Table 4: SINGULARITY-temporal results on text-to-video retrieval.

| Method | #PT | #Train Frame | MSRVTT | | | DiDeMo | | | ActivityNet Cap | | |
|---|---|---|---|---|---|---|---|---|---|---|---|
| | | | R1 | R5 | R10 | R1 | R5 | R10 | R1 | R5 | R10 |
| HERO (Li et al., 2020a) | 136M | 310 | 20.5 | 47.6 | 60.9 | - | - | - | - | - | - |
| MMT (Gabeur et al., 2020) | 136M | 1K/-/3K | 26.6 | 57.1 | 69.6 | - | - | - | 28.7 | 61.4 | 94.5 |
| ClipBERT (Lei et al., 2021) | 0.2M | 16/16/8 | 22.0 | 46.8 | 59.9 | 20.4 | 48.0 | 60.8 | 21.3 | 49.0 | 63.5 |
| VideoCLIP (Xu et al., 2021) | 136M | 960 | 30.9 | 55.4 | 66.8 | - | - | - | - | - | - |
| Frozen (Bain et al., 2021) | 5M | 4 | 31.0 | 59.5 | 70.5 | 31.0 | 59.8 | 72.4 | - | - | - |
| AlignPrompt (Li et al., 2021a) | 5M | 8 | 33.9 | 60.7 | 73.2 | 35.9 | 67.5 | 78.8 | - | - | - |
| CLIP4Clip (Luo et al., 2021) | 400M | 12/64/64 | 42.0 | 68.6 | 78.7 | 42.8 | 68.5 | 79.2 | 40.5 | 72.4 | 98.2 |
| SINGULARITY | 5M | 1 | 36.8 | 65.9 | 75.5 | 47.4 | 75.2 | 84.0 | 43.0 | 70.6 | 81.3 |
| SINGULARITY-temporal | 5M | 4 | 39.9 | 67.3 | 76.0 | 49.2 | 77.5 | 85.4 | 45.9 | 73.3 | 83.8 |
| SINGULARITY | 17M | 1 | 41.5 | 68.7 | 77 | **53.9** | 79.4 | 86.9 | 47.1 | 75.5 | 85.5 |
| SINGULARITY-temporal | 17M | 4 | **42.7** | **69.5** | **78.1** | 53.1 | **79.9** | **88.1** | **48.9** | **77.0** | **86.3** |

Table 5: SINGULARITY-temporal results on video question answering.

| Method | #PT | #Train Frame | MSRVTT-QA | ActivityNet-QA | MSRVTT-MC |
|---|---|---|---|---|---|
| ClipBERT (Lei et al., 2021) | 0.2M | 16 | 37.4 | - | 88.2 |
| AlignPrompt (Li et al., 2021a) | 5M | 16 | 42.1 | - | - |
| JustAsk (Yang et al., 2021) | 69M | 640 | 41.5 | 38.9 | - |
| MERLOT (Zellers et al., 2021) | 180M | 5 | 43.1 | 41.4 | 90.9 |
| VideoCLIP (Xu et al., 2021) | 136M | 960 | - | - | 92.1 |
| SINGULARITY | 5M | 1 | 42.7 | 41.8 | 92.0 |
| SINGULARITY-temporal | 5M | 4 | 43.3 | 43.4 | 92.0 |
| SINGULARITY | 17M | 1 | **43.5** | **43.1** | **92.1** |
| SINGULARITY-temporal | 17M | 4 | **43.9** | **44.1** | **93.7** |

**Zero-Shot Results.** In Table 6 we show zero-shot results of SINGULARITY for text-to-video retrieval. SINGULARITY achieves significantly better results compared to existing methods with a similar amount of pre-training data.

**Performance of Multiple Runs.** In Table 7 we show mean and standard deviation of 5 random runs, for text-to-video retrieval.

**Comparison on Inference Cost.** In Table 8, we compare the cost of various retrieval methods in terms of inference GFLOPs and the number of model parameters. Overall, SINGULARITY models have a similar amount of parameters and lower inference GFLOPs, with higher performance.

**Ablation Study on Training Objectives.** In Table 9, we study the effect of using different training objectives. We notice that using all objectives achieves the best performance. One interesting note is that, compared to (ITM+MLM), adding ITC loss (ITM+MLM+ITC) greatly improves retrieval performance on MSRVTT, but not ActivityNet QA. This makes sense as ITC is not applied on the multi-modal encoder which QA tasks may heavily rely on.

**Impact of Image Size.** In Figure 10 we study the impact of image size for downstream tasks. In general, a larger image size helps improve model performance, but the performance saturates at a certain size, e.g., the model performance saturates at around 336×336 for the 3 tasks. Note that our model performance with larger image sizes might suffer from the low resolution of the raw videos we have. For example, we are only able to get videos of resolution 320×240 for MSRVTT.

**Comparison on Image-Text tasks.** Since our model is pre-trained with single frames, it can be directly used for image-text tasks. In Table 11 we show image-text retrieval results on

Table 6: SINGULARITY zero-shot results on text-to-video retrieval.

| Method | #PT | #Train Frame | MSRVTT | | | DiDeMo | | | ActivityNet Cap | | |
|---|---|---|---|---|---|---|---|---|---|---|---|
| | | | R1 | R5 | R10 | R1 | R5 | R10 | R1 | R5 | R10 |
| VideoCLIP (Xu et al., 2021) | 137M | 1K | 10.4 | 22.2 | 30.0 | 16.6 | 46.9 | - | - | - | - |
| Frozen (Bain et al., 2021) | 5M | 4 | 18.7 | 39.5 | 51.6 | 21.1 | 46.0 | 56.2 | - | - | - |
| AlignPrompt (Li et al., 2021a) | 5M | 8 | 24.1 | 44.7 | 55.4 | 23.8 | 47.3 | 57.9 | - | - | - |
| CLIP-straight | 400M | 1 | 31.2 | 53.7 | 64.2 | - | - | - | - | - | - |
| BLIP | 130M | 1 | 43.3 | 65.6 | 74.7 | - | - | - | - | - | - |
| SINGULARITY | 5M | 1 | 28.4 | 50.2 | 59.5 | 36.9 | 61.1 | 69.3 | **30.8** | **55.9** | 66.3 |
| SINGULARITY | 17M | 1 | **34.0** | **56.7** | **66.7** | **37.1** | **61.7** | **69.9** | 30.6 | 55.6 | **66.9** |

Table 7: SINGULARITY results on text-to-video retrieval, with mean/std over 5 random runs. We show the results for the model pre-trained on the 17M corpus.

| Method | MSRVTT | | | DiDeMo | | | ActivityNet | | |
|---|---|---|---|---|---|---|---|---|---|
| | R1 | R5 | R10 | R1 | R5 | R10 | R1 | R5 | R10 |
| SINGULARITY | $42.1_{\pm0.5}$ | $69.3_{\pm0.4}$ | $78.1_{\pm0.7}$ | $53.3_{\pm1.0}$ | $78.7_{\pm1.3}$ | $86.3_{\pm1.5}$ | $47.0_{\pm0.5}$ | $75.7_{\pm0.3}$ | $85.3_{\pm0.3}$ |

Flickr30K Young et al. (2014) and COCO Chen et al. (2015). In Table 12 we show image question answering results on VQA Antol et al. (2015). We observe that SINGULARITY demonstrates competitive performance on the image-text tasks. As we still see a gap with state-of-the-art image-text models such as Li et al. (2022), one future direction is to adopt improved designs in these methods to further improve video-text task performance.

**Hyper-Parameters.** The hyper-parameters for our pre-training and downstream task fine-tuning are listed in Table 13 and Table 14. Note that we did not do an extensive hyper-parameter search, but mostly use the same hyper-parameters for different datasets under the same task, it is possible that better results can be achieved with more tuning.

## A.4 ADDITIONAL DATA DETAILS

**Statistics.** We show statistics of pre-training datasets in Table 15, and downstream datasets in Table 16.

**License.** We show dataset licenses in Table 17.

Table 8: Comparison of recent retrieval methods on inference GLOPs and #params. For brevity, we show DiDeMo retrieval performance with Average Recall (AvgR) – the average of R{1,5,10}.

| Method | #PT | Inference GFLOPs | #params | DiDeMo AvgR |
|---|---|---|---|---|
| Frozen (Bain et al., 2021) | 5M | 542 | 181M | 54.4 |
| AlignPrompt (Li et al., 2021a) | 5M | - | 231M | 60.7 |
| CLIP4Clip (Radford et al., 2021) | 400M | 1,121 | 164M | 63.5 |
| SINGULARITY | 5M | 451 | 202M | 68.9 |
| SINGULARITY-temporal | 5M | 485 | 209M | **70.7** |

Table 9: Ablation study on training objectives. The models are pre-trained on 2.5M WebVid video-text pairs for 10 epochs and are then fine-tuned.

| Objectives | MSRVTT Retrieval AvgR | ActivityNet-QA |
|---|---|---|
| ITM | 32.4 | 40.2 |
| ITM + MLM | 52.5 | 47.0 |
| ITM + ITC | 54.3 | 44.1 |
| ITM + MLM + ITC | 55.7 | 46.4 |

Table 10: Impact of Image Size. We fine-tune models from the same checkpoint, pre-trained with input image size 224×224. We show average recall (average of R@{1,5,10}) for retrieval tasks, and accuracy for the QA task.

| Image size | MSRVTT retrieval | DiDeMo retrieval | ActivityNet QA |
|---|---|---|---|
| 112 | 58.7 | 65.9 | 46.6 |
| 224 | 62.4 | **73.4** | 49.2 |
| 336 | **65.5** | **73.4** | 49.6 |
| 448 | 64.2 | 72.9 | **49.8** |

Table 11: Comparison to existing methods on image-text retrieval. We show results for both text retrieval (image-to-text retrieval, TR) and image retrieval (IR).

| Method | #PT | COCO (5K test) | | | | | | Flickr30K (1K test) | | | | | |
|---|---|---|---|---|---|---|---|---|---|---|---|---|---|
| | | TR | | | IR | | | TR | | | IR | | |
| | | R1 | R5 | R10 | R1 | R5 | R10 | R1 | R5 | R10 | R1 | R5 | R10 |
| ViLT (Kim et al., 2021) | 4M | 61.5 | 86.3 | 92.7 | 42.7 | 72.9 | 83.1 | 83.5 | 96.7 | 98.6 | 64.4 | 88.7 | 93.8 |
| UNITER (Chen et al., 2020) | 4M | 65.7 | 88.6 | 93.8 | 52.9 | 79.9 | 88.0 | 87.3 | 98.0 | 99.2 | 75.6 | 94.1 | 96.8 |
| OSCAR (Li et al., 2020b) | 4M | 70.0 | 91.1 | 95.5 | 54.0 | 80.8 | 88.5 | - | - | - | - | - | - |
| Frozen (Bain et al., 2021) | 5M | - | - | - | - | - | - | - | - | - | 61.0 | 87.5 | 92.7 |
| ALBEF (Li et al., 2021b) | 4M | 73.1 | 91.4 | 96.0 | 56.8 | 81.5 | 89.2 | 94.3 | 99.4 | 99.8 | 82.8 | 96.7 | 98.4 |
| ALBEF (Li et al., 2021b) | 14M | 77.6 | 94.3 | 97.2 | 60.7 | 84.3 | 90.5 | 95.9 | 99.8 | 100.0 | 85.6 | 97.5 | 98.9 |
| BLIP (Li et al., 2022) | 14M | 80.6 | 95.2 | 97.6 | 63.1 | 85.3 | 91.1 | 96.6 | 99.8 | 100.0 | 87.2 | 97.5 | 98.8 |
| BLIP (Li et al., 2022) | 129M | 81.9 | 95.4 | 97.8 | 64.3 | 85.7 | 91.5 | 97.3 | 99.9 | 100.0 | 87.3 | 97.6 | 98.9 |
| ALIGN (Jia et al., 2021) | 1.2B | 77.0 | 93.5 | 96.9 | 59.9 | 83.3 | 89.8 | 95.3 | 99.8 | 100.0 | 84.9 | 97.4 | 98.6 |
| SINGULARITY | 5M | 71.9 | 90.8 | 95.4 | 54.6 | 80.0 | 87.8 | 93.3 | 99.4 | 99.8 | 81.4 | 95.8 | 97.9 |
| SINGULARITY | 17M | 77.0 | 93.7 | 96.8 | 59.6 | 83.4 | 90.0 | 96.1 | 99.8 | 99.9 | 84.7 | 96.8 | 98.3 |

Table 12: Comparison to existing methods on VQA.

| Method | #PT | test-dev | test-std |
|---|---|---|---|
| ClipBERT (Lei et al., 2021) | 0.2M | 69.08 | 69.43 |
| ViLT (Kim et al., 2021) | 4M | 70.94 | - |
| VL-BART (Cho et al., 2021) | 0.2M | - | 71.30 |
| LXMERT (Tan & Bansal, 2019) | 4M | 72.42 | 72.54 |
| UNITER (Chen et al., 2020) | 4M | 72.70 | 72.91 |
| UNIMO (Li et al., 2021c) | 4M | 73.79 | 74.02 |
| OSCAR (Li et al., 2020b) | 4M | 73.16 | 73.44 |
| ALBEF (Li et al., 2021b) | 4M | 74.54 | 74.70 |
| ALBEF (Li et al., 2021b) | 14M | 75.84 | 76.04 |
| BLIP (Li et al., 2022) | 14M | 77.54 | 77.62 |
| BLIP (Li et al., 2022) | 129M | 78.24 | 78.17 |
| SINGULARITY | 5M | 70.30 | 70.53 |
| SINGULARITY | 17M | 73.13 | 73.27 |

Table 13: SINGULARITY hyper-parameters for pre-training, video QA, image QA and text-to-image retrieval. We only list a single value if all tasks share the same value. For SINGULARITY-temporal, we train with a similar setup, except that we set #training frames to be 4. In addition, for SINGULARITY-temporal 2nd stage pre-training, we also use a smaller batch size of 32 per GPU.

| config | pre-training | video QA | image QA | text-to-image retrieval |
|---|---|---|---|---|
| optimizer | AdamW (Loshchilov & Hutter, 2019) | | | |
| optimizer momentum | $\beta_1, \beta_2$=0.9,0.999 | | | |
| base learning rate | 1e-4 | 1e-5 | 1e-5 | 1e-5 |
| min learning rate | 1e-5 | 1e-6 | 1e-6 | 1e-6 |
| weight decay | 0.02 | | | |
| learning rate schedule | cosine decay (Loshchilov & Hutter, 2017) | | | |
| image size | 224 | 224 | 336 | 336 |
| image augmentation | random resize, crop, horizontal flip | | | |
| #training epochs | 10 | 10 | 5 | 10 (Flickr30K), 5 (COCO) |
| #warmup epochs | 1 | 0.5 | 0.5 | 0 |
| batch size x #GPUs | 128×3 | 32×1 | 64×4 | 64×2 |
| #training frames | | | 1 | |
| #inference frames | - | 12 | 1 | 1 |

Table 14: SINGULARITY hyper-parameters for text-to-video retrieval tasks. We only list a single value if all tasks share the same value. For SINGULARITY-temporal, we train it with a similar setup, except that we set #training frames to be 4.

| config | MSRVTT | DiDeMo | ActivityNet Captions | SSv2-template/label |
|---|---|---|---|---|
| optimizer | | | AdamW Loshchilov & Hutter (2019) | |
| optimizer momentum | | | $\beta_1, \beta_2$=0.9,0.999 | |
| base learning rate | 1e-5 | 1e-5 | 1e-5 | 1e-4 |
| min learning rate | 1e-6 | 1e-6 | 1e-6 | 1e-5 |
| weight decay | | | 0.02 | |
| learning rate schedule | | | cosine decay Loshchilov & Hutter (2017) | |
| image size | | | 224 | |
| image augmentation | | | random resize, crop, horizontal flip | |
| #training epochs | 5 | 10 | 10 | 10 |
| #warmup epochs | | | 0 | |
| batch size x #GPUs | 32x1 | 32x1 | 32x1 | 32x2 |
| #training frames | | | 1 | |
| #inference frames | 12 | 12 | 32 | 12 |

Table 15: Statistics of pre-training datasets. The average video length of WebVid is 18 seconds.

| Dataset | #image/video | #text | Type |
|---|---|---|---|
| COCO (Chen et al., 2015) | 113K | 567K | image |
| VG (Krishna et al., 2017b) | 100K | 768K | image |
| SBU (Ordonez et al., 2011) | 860K | 860K | image |
| CC3M (Sharma et al., 2018) | 2.95M | 2.95M | image |
| CC12M (Changpinyo et al., 2021) | 10.77M | 10.77M | image |
| WebVid (Bain et al., 2021) | 2.49M | 2.49M | video |
| 5M corpus = CC3M+WebVid | 5.44M | 5.44M | video+image |
| 17M corpus = 5M+COCO+VG+SBU+CC12M | 17.28M | 18.41M | video+image |

Table 16: Statistics of downstream datasets.

| Dataset | #video | | | #text | | | Avg Video |
|---|---|---|---|---|---|---|---|
| | Train | Val | Test | Train | Val | Test | Length (s) |
| *Text-to-Video Retrieval* | | | | | | | |
| ActivityNet Cap (Krishna et al., 2017a) | 10,009 | - | 4,917 | 10,009 | - | 4,917 | 180 |
| DiDeMo (Anne Hendricks et al., 2017) | 8,394 | 1,065 | 1,003 | 8,394 | 1,065 | 1,003 | 29.3 |
| MSRVTT (Xu et al., 2016) | 7,010 | - | 1,000 | 140,200 | | 1,000 | 15 |
| SSV2-Template (Goyal et al., 2017a) | 168,913 | - | 2,088 | 174 | - | 174 | 4 |
| SSV2-Label (Goyal et al., 2017a) | 168,913 | - | 2,088 | 109,968 | - | 1,989 | 4 |
| *Video Question Answering* | | | | | | | |
| MSRVTT-QA (Xu et al., 2017) | 6,513 | 497 | 2,990 | 158,581 | 12,278 | 72,821 | 15 |
| ActivityNet-QA (Yu et al., 2019) | 3,200 | 1,800 | 800 | 32,000 | 18,000 | 8,000 | 180 |
| MSRVTT-MC (Yu et al., 2018) | 7,010 | - | 2,990 | 140,200 | | 14,950 | 15 |

Table 17: Dataset licenses.

| Dataset | License |
| --- | --- |
| COCO (Chen et al., 2015) | CC BY 4.0, Flickr Terms of Use |
| VG (Krishna et al., 2017b) | CC BY 4.0 |
| SBU (Ordonez et al., 2011) | Flickr Terms of Use |
| CC3M (Sharma et al., 2018) | CC3M License |
| CC12M (Changpinyo et al., 2021) | CC12M License |
| WebVid (Bain et al., 2021) | Exceptions to Copyright |
| ActivityNet Captions (Krishna et al., 2017a) | Fair Use |
| DiDeMo (Anne Hendricks et al., 2017) | BSD-2-Clause, Creative Commons |
| MSRVTT (Xu et al., 2016) | unknown |
| SSV2-Template (Goyal et al., 2017a) | SSv2 License |
| SSV2-Label (Goyal et al., 2017a) | SSv2 License |
| MSRVTT-QA (Xu et al., 2017) | MIT |
| ActivityNet-QA (Yu et al., 2019) | Apache |
| MSRVTT-MC (Yu et al., 2018) | unknown |

