# OpenReview forum: "Revealing Single Frame Bias for Video-and-Language Learning"
_ICLR.cc/2023/Conference — Submitted to ICLR 2023_

### Official Review · Reviewer_VHWz · 2022-10-23

**Confidence:** 5
**Correctness:** 2
**Technical Novelty And Significance:** 3
**Empirical Novelty And Significance:** 3
**Recommendation:** 5

**Clarity, Quality, Novelty And Reproducibility:**

This paper is clearly written and easy to follow. The author also provides the code of the paper.

**Strength And Weaknesses:**

Strengths:
1. The single-frame training strategy is highly efficient while effective, which only needs to train one frame during pre-training and fine-tuning.
2. The proposed model achieves state-of-the-art among the video-text retrieval tasks on VideoQA tasks.
3. This paper is clearly written and easy to follow.

Weaknesses:
1. The single-frame pre-training strategy (see Figure 6) is not fully convincing to me. Although the author claim that “when pre-trained on a sufficient amount of data, the performance of models trained with single frames might be very close to models trained with multiple frames”. However, there only exists 2.5M video-text data among the full 17M pre-training dataset. I think the reason is that with the growth of the #PT images/videos, the large-scale image-text dataset is enough for the model to achieve a good performance on downstream tasks (Note that a lot of recent works [1] have demonstrated that a pure image-text dataset is enough for video-text downstream tasks).
2. Meanwhile, with the growth of the pre-training video-text dataset, there inevitably exists more noisy information and irrelevant frames in the video. Will the single-frame strategy still work on these large-scale web video datasets? I think an additional experiment on a pure large-scale web video dataset (instead of a pre-training dataset mix of videos and large portion images) will further enhance the reliability of this paper (e.g., HowTo100M).
3. The proposed early-fusion strategy is simple and effective, which I do appreciate. However, compared with late-fusion, early-fusion strategy seems to need to input an Nx longer frame feature sequence into the multi-modal encoder. And this may bring an Nx higher memory/inference times cost compared with the late-fusion strategy. The author could further provide a memory/time cost comparison in Figure 4.


**Summary Of The Paper:**

This paper explores single-frame training for video-and language tasks. While simple, this approach can achieve state-of-the-art performance on a range of datasets. This paper also proposes an early-fusion strategy at inference which boosts performance.

**Summary Of The Review:**

Although a lot of paper has proposed the single-frame pre-training strategy (on the pure image-text pre-training dataset), the single-frame finetuning strategy is still novel to me. My main concerns are the reliability of the single-frame strategy on the pure large-scale web video datasets and the memory/time cost of the early-fusion strategy. I will raise my score if these concerns are addressed.

---

> ### Author Response · Authors · 2022-11-19
> **Response to Reviewer VHWz**
>
> > Meanwhile, with the growth of the pre-training video-text dataset, there inevitably exists more noisy information and irrelevant frames in the video. Will the single-frame strategy still work on these large-scale web video datasets? I think an additional experiment on a pure large-scale web video dataset (instead of a pre-training dataset mix of videos and large portion images) will further enhance the reliability of this paper (e.g., HowTo100M).
>
> Thanks for your suggestion. We pre-train our single-frame model on the large-scale web video dataset WebVid-10M with 10M videos, and the results are shown below. For comparison, we also show model results without pre-training; & pre-trained on 2.5M WebVid-2M videos. From this table, we notice that when increasing pre-training video size, 0 → 2.5M → 10M, model performance consistently increases across the 4 datasets, e.g., for MSRVTT retrieval R1: 24.89 → 30.82 → 35.80. We also notice that model results with pre-training on 10M videos are close to that pre-trained on 2.5M videos + 3M images. Overall, this demonstrates that our single-frame model does benefit from more pre-training videos; and that pre-training with extra image data is very helpful.
>
> | Method | #PT Videos | #PT Images | #Train | Retrieval R1 |  |  | QA accuracy |
> |---|---|---|---|---|---|---|---|
> |  |  |  | Frames | SSv2-label | MSRVTT | DiDeMo | ActivityNet-QA |
> | Frozen | 2.5M | 3M | 4 | - | 31.00 | 31.00 | - |
> | AlignPrompt | 2.5M | 3M | 4 | 39.32 | 33.90 | 35.90 | - |
> | CLIP4Clip | 0 | 400M | 12 | 43.14 | 42.00 | 42.80 | - |
> | JustAsk | 0 | 69M | 640 | - | - | - | 41.50 |
> | MERLOT | 0 | 180M | 5 | - | - | - | 43.10 |
> |  |  |  |  |  |  |  |  |
> | Singularity | 0 | 0 | 1 | 24.89 | 15.70 | 8.47 | 36.25 |
> | Singularity | 2.5M | 0 | 1 | 30.82 | 30.80 | 32.70 | 39.17 |
> | **Singularity** | **10M** | 0 | 1 | 35.80 | 36.30 | 45.76 | 40.76 |
> | Singularity | 2.5M | 3M | 1 | 36.40 | 36.80 | 47.40 | 41.80 |
>
>
> > The proposed early-fusion strategy is simple and effective, which I do appreciate. However, compared with late-fusion, early-fusion strategy seems to need to input an Nx longer frame feature sequence into the multi-modal encoder. And this may bring an Nx higher memory/inference times cost compared with the late-fusion strategy. The author could further provide a memory/time cost comparison in Figure 4.
>
> Thanks for this suggestion. We added detailed memory and time cost of the fusion methods in Appendix Section A.1 (title highlighted in red)., of the updated paper PDF. Thanks to the use of cross-attention, theoretically, our early-fusion strategy only has linear computation cost, which is comparable to the late-fusion strategies. In Section A.1 Figure 8, we compared the performance, time and memory cost of early-fusion (`concat`) and late-fusion (`lse`) strategy. We show that our early-fusion strategy runs faster than late-fusion, while also achieve better performance for both MSRVTT retrieval and ActivityNet-QA tasks. For memory, early fusion has lower cost than late fusion on ActivityNet-QA, but higher cost on MSRVTT retrieval task. Overall, the early fusion approach is preferred in most cases due to its better accuracy and faster run time.
>
>
> > The single-frame pre-training strategy (see Figure 6) is not fully convincing to me. Although the author claim that “when pre-trained on a sufficient amount of data, the performance of models trained with single frames might be very close to models trained with multiple frames”. However, there only exists 2.5M video-text data among the full 17M pre-training dataset. I think the reason is that with the growth of the #PT images/videos, the large-scale image-text dataset is enough for the model to achieve a good performance on downstream tasks (Note that a lot of recent works [1] have demonstrated that a pure image-text dataset is enough for video-text downstream tasks).
>
> Thanks. We fully agree with your suggestion that doing pre-training on a larger scale video-text data will better support the claim we have in Figure 6. We plan to conduct experiments on the 10M video-text data from WebVid-10M to further strength the claim. Due to limited time and GPU resources, we are unable to finish the training of the temporal model (it takes more than 2 weeks on WebVid-10M). We are working on these experiments and will add the results in our final version.

---

> > ### Comment · Reviewer_VHWz · 2022-11-22
> > **Further comments**
> >
> > Thanks for your feedback. “We also notice that model results with pre-training on 10M videos are close to that pre-trained on 2.5M videos + 3M images.” It seems that pre-training on additional 7.5M (2.5x larger) videos is even worse than 3M images. I regret to say that the single-frame strategy seems not to work on these large-scale video datasets. And the superior performance in this paper seems to come from a large portion image-text dataset instead of "the single-frame strategy". Therefore, I keep my ratings.

---

> > > ### Author Response · Authors · 2022-11-22
> > > **Response to further comments**
> > >
> > > Thanks for your reply! The phenomenon "*It seems that pre-training on additional 7.5M (2.5x larger) videos is even worse than 3M images.*"  is expected. Randomly sample a single frame from a whole video and pair it with the original video caption can be seen as very noisy image-text pair. This also matches your intuition "*there inevitably exists more noisy information and irrelevant frames in the video*".
> > > While on the other hand, the web image-text pairs can be seen as much better aligned pairs without random frame sample noise, thus definitely is more powerful for training our single-frame model.
> > >
> > > Meanwhile, as you mentioned, "*Although a lot of paper has proposed the single-frame pre-training strategy (on the pure image-text pre-training dataset), the single-frame finetuning strategy is still novel to me.*", the main focus of this paper is that we found after large-scale single-frame pre-training and a good inference ensemble frame method, we are able to do **single-frame finetuning** on downstream tasks and achieve strong performance compared to methods that are finetuned on multiple frames. The webvid-10M experiment is only about using video-text data for single-frame pre-training, and we showed as more videos are added (2.5M --> 10M), our model is able to achieve better performance. This should answer your question "*Will the single-frame strategy still work on these large-scale web video datasets? *", and the answer is **Yes**.
> > >
> > > As for your last point "*And the superior performance in this paper seems to come from a large portion image-text dataset instead of "the single-frame strategy*", the best compared method **AlignPromot** is pre-trained on the same set of 3M image-text + 2.5M video-text pairs as our singularity 5M model, but achieves lower performance on existing datasets, e.g., 33.90 vs. 36.80 for MSRVTT retrieval, and  35.9 vs. 47.4. The same conclusion holds for the **Frozen** method. As the two other methods use the same pre-training image-text + video-text data, it is not reasonable to describe our method that "*the superior performance in this paper seems to come from a large portion image-text dataset instead of "the single-frame strategy"*.
> > >
> > > Please let us know if any further clarification is needed. Thanks again!

---

### Official Review · Reviewer_DUem · 2022-10-24

**Confidence:** 4
**Correctness:** 4
**Technical Novelty And Significance:** 2
**Empirical Novelty And Significance:** 3
**Recommendation:** 6

**Clarity, Quality, Novelty And Reproducibility:**

The paper is clear and well written. It has details needed to reproduce the results. It isn't really novel, but has important findings. The proposed new benchmark is interesting and valuable, however the new benchmark is totally clear how useful it is.

**Strength And Weaknesses:**

The paper is well written and easy to understand. The experiments are well done. The finding that many current tasks can be done with a single frame training is interesting and useful. For the proposed task, it would be good to add more details. For example, how many unique labels are there?  It also be valuable to evaluate more existing methods on this benchmark to better understand how existing approachs perform on it.

Minor details:
The bolding is misleading in Tables 1 and 2.
Should use citep command for references to make it easier to read.

**Summary Of The Paper:**

This paper studies a video-language model trained using only a single frame, finding that is can do quite well on many existing benchmarks. The paper then proposes new benchmarks on something-something, where more temporal understanding is required.

**Summary Of The Review:**

The paper studies an important and interesting problem, finding that single frames are good enough for many video language tasks and proposes a solution to it.

---

> ### Author Response · Authors · 2022-11-19
> **Response to Reviewer DUem**
>
> > The paper is well written and easy to understand. The experiments are well done. The finding that many current tasks can be done with a single frame training is interesting and useful. For the proposed task, it would be good to add more details. For example, how many unique labels are there?
>
> In Table 16 statistics of downstream datasets, we show that, for testing, SSv2-template dataset has 174 unique text queries with 2,088 unique videos; SSv2-label has 1989 unique text queries within the same 2,088 unique videos as in SSv2-template. We will release all the data and evaluation code we used for the new benchmark for easy reproduction.
>
> >  It also be valuable to evaluate more existing methods on this benchmark to better understand how existing approachs perform on it.
>
> Thanks! In this work, we evaluate the two open-sourced and representative methods frozen and CLIP4Clip on the new benchmark. Per your request, we also added one more recent strong method, AlignPrompt, which uses a TimeSformer model for temporal video encoding. The results are shown below (we will add the results to the final paper):
>
> | Method | #PT  | #Train  | SSv2-label |  |  | SSv2-template |  |  |
> |---|---|---|---|---|---|---|---|---|
> |  |  | Frame | R1 | R5 | R10 | R1 | R5 | R10 |
> | Frozen | 5M | 4 | - | - | - | 52.9 | 94.8 | 99.4 |
> | **AlignPrompt** | 5M | 4 | 39.3 | 68.7 | 79.1 | 64.4 | 97.7 | 99.4 |
> | CLIP4Clip | 400M | 12 | 43.1 | 71.4 | 80.7 | 77.0 | 96.6 | 98.3 |
> | Singularity | 5M | 1 | 36.4 | 64.9 | 75.4 | 42.0 | 86.2 | 94.3 |
> | Singularity-temporal | 5M | 4 | 44.1 | 73.5 | 82.2 | 77.0 | 98.9 | 99.4 |
> | Singularity-temporal | 17M | 4 | 47.4 | 75.9 | 84.0 | 77.6 | 96.0 | 98.9 |
>
> The conclusions from our paper remain – though existing temporal methods underperform our single-frame Singularity model using the same 5M data for most existing datasets (See main paper Table 1, 2), but they are stronger on the newly proposed SSv2 datasets, showing that temporal modeling is more important in the SSv2 datasets than existing datasets.
>
> > Minor details: The bolding is misleading in Tables 1 and 2. Should use citep command for references to make it easier to read.
>
> Thanks! We have replaced `\cite` with `\citep` in tables 1,2 following your suggestion.

---

### Official Review · Reviewer_kab7 · 2022-10-27

**Confidence:** 4
**Correctness:** 3
**Technical Novelty And Significance:** 2
**Empirical Novelty And Significance:** 3
**Recommendation:** 3

**Clarity, Quality, Novelty And Reproducibility:**

The paper is clearly written for the most part. There is limited novelty in this work.

**Strength And Weaknesses:**

**Strengths**

-- The empirical studies presented in the work are insightful and will help propel more understanding in this relatively new domain of text-video understanding.


**Weaknesses**

-- While the empirical studies are certainly insightful, the paper's contributions are limited to just that. There are no other significant technical contributions.

-- While the training is done with single frame, inference is still conducted using multiple sampled frames, which makes the setup a bit hard to fairly evaluate.

-- The claim in the paper that existing datasets in the space (ActivityNet, MSRVTT) do not require any temporal understanding is only partially correct. Given that R1 results are still ~50%, it means that these are not solved datasets. The results point more towards the limitation of existing methods in fully learning/exploiting the temporal information in the videos rather a strict limitation of the datasets.

-- The two new proposed datasets SSv2-{label|template} also have very high (and comparable) R1 scores with just sampling single frame which means those 2 new datasets are roughly comparable with other existing datasets.

-- Overall, since the technical contribution in the paper is quite limited, the Sec 5 (Analysis) does not present any interesting studies and felt forced in the paper.

**Summary Of The Paper:**

This paper studies the domain of video-and-language tasks and analyses the methods and datasets in the space. Through empirical studies, authors show that using a single frame (randomly sampled) from the entire video is enough to reach similar performance as existing SOTA methods on these datasets. The claim is that such tasks do not need temporal understanding, merely static appearance/image understanding. Further, two new datasets are proposed in this space that require more temporal understanding.

**Summary Of The Review:**

Overall, not very convinced the paper is ready in its current form, especially as a long-form paper. The technical contributions are limited. The empirical studies do point to limitations of existing methods (and datasets to some extent) but do not propose any effective solutions to overcome such limitations.

---

> ### Author Response · Authors · 2022-11-19
> **Response to Reviewer kab7**
>
> > While the empirical studies are certainly insightful, the paper's contributions are limited to just that. There are no other significant technical contributions
>
> We do not claim novelty in architecture, etc, but focus on studying how we could achieve strong single-frame performance with such architecture:  large-scale pre-training & an early fusion strategy. Our contributions are highlighted by other reviewers, e.g., `Reviewer VHWz`: *“Although a lot of paper has proposed the single-frame pre-training strategy (on the pure image-text pre-training dataset), the single-frame finetuning strategy is still novel to me.”*. `Reviewer pLom`: *“The proposed approach is simple and effective, and the authors prove that with the simple, single-frame model, the video-language models can already achieve good performance.”* `Reviewer DUem` mentioned that *“The paper is well written and easy to understand. The experiments are well done. The finding that many current tasks can be done with a single frame training is interesting and useful.”*
>
> > While the training is done with single frame, inference is still conducted using multiple sampled frames, which makes the setup a bit hard to fairly evaluate.
>
> During inference, existing methods also use multiple frames (the same #frames as in their training). Below we compare #inference frames of our methods with the best baseline CLIP4Clip. For *#Inference Frames*, *“12/64/64”* means 12, 64, 64 frames are used during inference for the 3 datasets respectively. We observe that we are using the same or fewer frames during inference, e.g., for DiDeMo, CLIP4Clip uses 64 frames while our 5M model use only 12 frames, but it still achieves 4.6 R1 gain (47.4 vs. 42.8). This shows that our approach works even under unfair comparison. This is especially true considering that we are using fewer amount of pre-training data (5M vs. 400M).
>
> | Method | #PT | #Train Frames | #Inference Frames | MSRVTT | DiDeMo | ActivityNet Cap |
> |---|---|---|---|---|---|---|
> | CLIP4Clip | 400M | 12/64/64 | 12/64/64 | 42 | 42.8 | 40.5 |
> | Singularity | 5M | 1 | 12/12/32 | 36.8 | 47.4 | 43 |
> | Singularity | 17M | 1 | 12/12/32 | 41.5 | 53.9 | 47.1 |
>
> > The claim in the paper that existing datasets in the space (ActivityNet, MSRVTT) do not require any temporal understanding is only partially correct. Given that R1 results are still ~50%, it means that these are not solved datasets. The results point more towards the limitation of existing methods in fully learning/exploiting the temporal information in the videos rather a strict limitation of the datasets.
>
> We believe that you have misunderstood our claim. In the abstract, we mentioned *“This result reveals the existence of a strong “static appearance bias” in popular video-and-language datasets.”*, and in the introduction, *“More importantly, this strong single-frame performance reveals that the current evaluation is biased towards still objects, scenes, etc., while the temporal dynamics seem negligible, which in fact should be important for “true” video-language understanding.”* As you can see from these two quotes, we only claim that current video-text datasets are biased toward static objects/scenes/etc, while temporal information seems less useful; we don’t claim that they “do not require **any** temporal understanding”. We fully agree that, in the future, a more advanced static+temporal model will achieve the best overall performance.
>
> > The two new proposed datasets SSv2-{label|template} also have very high (and comparable) R1 scores with just sampling single frame which means those 2 new datasets are roughly comparable with other existing datasets.
>
> We respectfully disagree, as the new SSv2-{label|template} datasets are quite different from existing datasets. In the table below, we compare Singularity models with existing methods on various retrieval datasets, under the R1 metric. When only using our 1-frame model (Singularity), the two new SSv2 datasets have quite low results compared to existing methods. E.g., when comparing to 12-frame CLIP4Clip on SSv2-template, 1-frame Singularity has a 35.0 lower R1 (42.0 vs. 77.0), this is very different from existing datasets such as DiDeMo, where 1-frame Singularity has 4.6 higher R1 (47.4 vs. 42.8). This actually demonstrates that single-frame model works well for existing datasets, but not for SSv2 datasets. Thus, SSv2 datasets are very different from existing ones. Also, the Singularity-temporal is a 4-frame temporal model, the fact that it works much better than 1-frame model also shows temporal understanding is important for SSv2-{label|template}.
>
> | Method | #PT | #Train Frames | SSv2-label | SSv2-template | MSRVTT | DiDeMo | ActivityNet Cap |
> |---|---|---|---|---|---|---|---|
> | Frozen | 5M | 4 | - | 52.9 | 31 | 31 | - |
> | CLIP4Clip | 400M | 12 | 43.1 | 77 | 42 | 42.8 | 40.5 |
> | Singularity | 5M | 1 | 36.4 | 42 | 36.8 | 47.4 | 43 |
> | Singularity-temporal | 5M | 4 | 44.1 | 77 | 39.9 | 49.2 | 45.9 |

---

> > ### Author Response · Authors · 2022-11-19
> > **Extended Response to Reviewer kab7**
> >
> > > Overall, since the technical contribution in the paper is quite limited, the Sec 5 (Analysis) does not present any interesting studies and felt forced in the paper.
> >
> > Our analysis in Sec. 5 does not focus on studying new architectures, but instead how we can achieve state-of-the-art performance on video-text datasets using a single-frame approach. In the section, we show that (1) large-scale pre-training on image-text and video-text datasets; and (2) a good early fusion strategy are the two important factors. We also use quantitative examples to demonstrate why early fusion is better than late fusion for fusing information from multiple frames at inference. All of these are not clearly studied in previous work and are useful for future work in exploring this direction.
> >
> > > The empirical studies do point to limitations of existing methods (and datasets to some extent) but do not propose any effective solutions to overcome such limitations.
> >
> > We recognize that the current evaluation is biased toward static appearance recognition, and propose the new SSv2 retrieval datasets to help the community better evaluate models’ temporal understanding ability.

---

### Official Review · Reviewer_pLom · 2022-11-01

**Confidence:** 4
**Correctness:** 4
**Technical Novelty And Significance:** 2
**Empirical Novelty And Significance:** 2
**Recommendation:** 3

**Clarity, Quality, Novelty And Reproducibility:**

The paper is clearly written. The novelty is limited. The authors provide details of the proposed method to reproduce the results.

**Strength And Weaknesses:**

Strengths:
1. The paper is clearly written and easy to follow.
2. It reveals the single-frame bias for current video-language datasets.
3. The proposed approach is simple and effective, and the authors prove that with the simple, single-frame model, the video-language models can already achieve good performance.

Weaknesses:
1. The novelty is limited. The single-frame model applies the basic image-language architectural design with self-attention and cross-attention.
2. The single-frame bias for current video datasets is not new. Researchers have realized that most video datasets (except for SSv2) do not require much temporal information for understanding the contents.
3. The SSv2 retrieval tasks are only naive extensions of the SSv2 dataset.

**Summary Of The Paper:**

This paper investigates whether using multiple frames for training is necessary for video-language downstream tasks. The authors propose a single-frame framework for video-language understanding. Results indicate that with large-scale pre-training and a proper frame ensemble strategy at inference time, a single-frame trained model that does not consider temporal information can achieve better performance than existing methods that use multiple frames for training. This result reveals strong static appearance bais in current video-langauge datasets. To allow for a more comprehensive evaluation of video-and-language models, the authors propose two new retrieval tasks based on existing fine-grained action recognition datasets that encourage temporal modeling.

**Summary Of The Review:**

My main concern is the limited novelty, as it has been investigated by previous works that most video datasets do not need too much temporal information for understanding. The proposed single-frame architecture is a basic structure with self-attention and cross-attention, and the novelty is limited.

---

> ### Author Response · Authors · 2022-11-19
> **Response to Reviewer pLom**
>
> > The novelty is limited. The single-frame model applies the basic image-language architectural design with self-attention and cross-attention.
>
> We do not claim novelty in this architecture itself, but focus on studying how we are able to achieve strong single-frame performance with such architecture, where our solution is large-scale pre-training and an early fusion strategy. Also note that our use of cross-attention for early fusion of video frames is also new to the community since previous work mostly aggregates frames using mean-pooling like in ClipBERT or space-time attention as in Frozen. Our contributions are highlighted by other reviewers, for example, `Reviewer VHWz` mentioned that “Although a lot of paper has proposed the single-frame pre-training strategy (on the pure image-text pre-training dataset), the single-frame finetuning strategy is still novel to me.” and “The proposed early-fusion strategy is simple and effective”, `Reviewer kab7` mentioned that “The empirical studies presented in the work are insightful and will help propel more understanding in this relatively new domain of text-video understanding.”
>
> > The single-frame bias for current video datasets is not new. Researchers have realized that most video datasets (except for SSv2) do not require much temporal information for understanding the contents.
>
> To the best of our knowledge, this bias has not yet been studied in the video-text domain, especially under the current standard pre-training + fine-tuning setup. While people have shown single-frame bias is true in some pure video datasets, it remains unclear whether it holds for video-text datasets, where text signals are added. It is important that we prove it using extensive experiments. We are also happy to discuss any related work we missed in the video-text domain if the reviewer can kindly point these to us.
>
> > The SSv2 retrieval tasks are only naive extensions of the SSv2 dataset.
>
> Thanks! We agree the creation of the SSv2 retrieval datasets is relatively simple extension, however, it is crucial that these benchmarks are established so that future work could utilize them and compare their results. In recognizing that SSv2 retrieval datasets have different behaviors (more temporal) compared to most existing video-text datasets, though simple, its importance should not be underrated. Meanwhile, also note that our focus is to study the static appearance bias in existing datasets and how to achieve state-of-the-art performance with a single frame trained model.

---

> > ### Comment · Reviewer_pLom · 2022-11-25
> > **Thank authors for the rebuttal**
> >
> > Thank the authors for the rebuttal. However, I still lean towards rejecting this paper because of its limited contributions.

---

> > > ### Author Response · Authors · 2022-11-25
> > > **Response to reviewer pLom reply.**
> > >
> > > Thanks for your reply! Novelty is often too vague and subjective, we would appreciate it if you could instead consider how our work could inspire and help future research in this field.

---

### Decision · Program_Chairs · 2023-01-20

**Decision:**

Reject

**Justification For Why Not Higher Score:**

The draft in its current form, with its claimed novelty, is not above the acceptance threshold of ICLR. I suggest rejecting.

**Justification For Why Not Lower Score:**

N/A

**Metareview: Summary, Strengths And Weaknesses:**

This paper studies the domain of video-and-language tasks and analyses the methods and datasets in the space. Through empirical studies, authors show that using a single frame (randomly sampled) from the entire video is enough to reach similar performance as existing SOTA methods on these datasets. The claim is that such tasks do not need temporal understanding, merely static appearance/image understanding. Further, two new datasets are proposed in this space that require more temporal understanding.

- I agree with reviewers' point the novelty is limited. The single-frame model applies the basic image-language architectural design with self-attention and cross-attention.
- Lack of significant technical contributions, and the experimental setting is hard to fairly evaluate compared with other methods.
-  the superior performance in this paper seems to come from a large portion image-text dataset instead of "the single-frame strategy".

The draft in its current form, with its claimed novelty, is not above the acceptance threshold of ICLR. I suggest rejecting.

**Summary Of Ac-Reviewer Meeting:**

N/A